# The Impact of Mitochondrial Fission-Stimulated ROS Production on Pro-Apoptotic Chemotherapy

**DOI:** 10.3390/biology10010033

**Published:** 2021-01-06

**Authors:** Jan Ježek, Katrina F. Cooper, Randy Strich

**Affiliations:** 1The Wellcome Trust/Gurdon Cancer Research Institute, University of Cambridge, Cambridge CB2 1QN, UK; 2Department of Genetics, University of Cambridge, Cambridge CB2 1QN, UK; 3Department of Molecular Biology, School of Osteopathic Medicine, Rowan University, Stratford, NJ 08084, USA; cooperka@rowan.edu (K.F.C.); strichra@rowan.edu (R.S.)

**Keywords:** mitochondria, reactive oxygen species, cancer, chemotherapy, oxidative stress, stress signaling, mitochondrial dynamics, mitophagy, apoptosis, cyclin C

## Abstract

**Simple Summary:**

Mitochondria are the core energy-generating units found within a cell. In addition, mitochondria harbor molecular factors that are essential, upon their release from these organelles, for triggering cell suicide program or apoptosis. Recent research has pointed to the critical role that the mitochondrial shape, which is dynamically flexible rather than rigid, plays in regulating both, bioenergetics metabolism and programmed cell death. Given that activating apoptosis specifically in tumor cells can be an advantage for eradicating cancer by chemotherapy, we address the simple idea of whether pharmacological stimulation of mitochondrial dynamics can benefit cancer patients with solid tumors. We propose a model, in which mitochondrial fragmented phenotype and mitochondrial reactive oxygen species (ROS) production are interconnected within a self-propagating cycle that relies for its function on nuclear stress signaling pathways. We conclude that manipulation of mitochondrial dynamics may be at the heart of chemotherapeutic approaches targeting cancers with elevated oxidative stress.

**Abstract:**

Cancer is one of the world’s deadliest afflictions. Despite recent advances in diagnostic and surgical technologies, as well as improved treatments of some individual tumor types, there is currently no universal cure to prevent or impede the uncontrolled proliferation of malignant cells. Targeting tumors by inducing apoptosis is one of the pillars of cancer treatment. Changes in mitochondrial morphology precede intrinsic apoptosis, but mitochondrial dynamics has only recently been recognized as a viable pharmacological target. In many cancers, oncogenic transformation is accompanied by accumulation of elevated cellular levels of ROS leading to redox imbalance. Hence, a common chemotherapeutic strategy against such tumor types involves deploying pro-oxidant agents to increase ROS levels above an apoptotic death-inducing threshold. The aim of this chapter is to investigate the benefit of stimulating mitochondrial fission-dependent production of ROS for enhanced killing of solid tumors. The main question to be addressed is whether a sudden and abrupt change in mitochondrial shape toward the fragmented phenotype can be pharmacologically harnessed to trigger a burst of mitochondrial ROS sufficient to initiate apoptosis specifically in cancer cells but not in non-transformed healthy tissues.

## 1. Introduction

During oxidative metabolism, incomplete reduction of oxygen leads to the formation of highly active compounds known as reactive oxygen species (ROS) [1]. ROS at physiological levels serve as intra- and extracellular signaling molecules but prolonged or excessive production of ROS can lead to the damage of proteins, DNA, lipids, or other cellular components [2]. Depending on its severity and nature, oxidative stress can activate pro-survival pathways such as mitophagy or triggering of regulated cell death (RCD) responses such as intrinsic apoptosis [3]. One of the common features of stress-induced mitophagy and apoptosis is increased fragmentation of the mitochondrial network. The ability of mitochondria to sense and dynamically respond to oxidative changes suggests the presence of redox-sensing signaling pathways that stimulate the mitochondrial fission machinery [4]. In addition, the critical decision whether to repair oxidative damage and survive, or undergo RCD, depends also on the detoxification and scavenging capacity of antioxidant mechanisms controlling cellular redox homeostasis [5,6]. Given that evasion of apoptosis is one of the hallmarks of cancer progression, it is of paramount importance to grasp a detailed molecular understanding of how mitochondrial dynamics and redox homeostasis modulate the cell’s decision to survive or undergo cell death [7]. On one hand, cancer cells produce excessive amounts of ROS as a byproduct of their reprogrammed metabolism to sustain increased metabolic needs for tumor growth [8]. On the other hand, elevated oxidative status may represent an Achilles’ heel for some cancer types as it may be more feasible to target these tumors with pro-oxidant chemotherapy [9]. The following sections review the basic concepts of mitochondrial dynamics, mitochondrial quality control pathways, redox biology, and mitochondrial-dependent apoptosis in the context of carcinogenesis.

## 2. Stress-Induced Signaling Directs Cell Fate Decisions

### 2.1. Mitochondria Are Highly Dynamic Organelles

Mitochondria form a highly flexible and interconnected network, in which individual units undergo frequent events of fission and fusion (Appendix A) [10]. Enzymes responsible for mediating the highly conserved process of mitochondrial dynamics involve large fission and fusion guanosine 5′-triphosphatases (GTPases) such as dynamin-related protein 1 (Drp1) or dynamin 2 (Dnm2) and mitofusins Mfn1 or Mfn2, respectively [11]. In addition, inner mitochondrial membrane (IMM) fusion is facilitated by the optic atrophy 1 (Opa1) GTPase [12]. Mitochondrial fusion plays an indispensable role in maintaining the homogeneity of the mitochondrial matrix content across the whole mitochondrial network. Namely, dynamic equilibrium in mitochondrial DNA (mtDNA) content may help to equally distribute the repertoire of different mtDNA variants throughout a heteroplasmic cell. The process of mitochondrial fission is thought to be initiated by association with the endoplasmic reticulum (ER), which pre-constricts the mitochondrial filament [13]. Upon stress stimulus, Drp1 is recruited to pre-constricted sites at the outer mitochondrial membrane (OMM) through interaction with OMM receptors such as mitochondrial fission factor (Mff), mitochondrial fission 1 (Fis1), or mitochondrial dynamic proteins 49 and 51 (MiD49 and MiD51) [14,15,16]. Following OMM binding and Drp1 oligomerization into spiral-shaped filaments wrapping around the pre-constricted mitochondrial tubule, GTP hydrolysis induces conformational changes that result in Drp1 ring closure and mitochondrial constriction [11,17]. Dnm2 is then recruited to pre-constricted sites to complete the scission of both OMM and IMM [18]. Stress-induced mitochondrial fission is regulated by posttranslational modification of Drp1 such as phosphorylation, ubiquitination, SUMOylation, *S*-nitrosylation, *O*-GlcNAcylation, and sulfenylation [10]. For example, phosphorylation of Drp1 at Ser616 and Ser637 has stimulatory and inhibitory effects on its activity, respectively [18]. Lastly, but not less importantly, mitochondrial fragmentation occurs during mitosis to facilitate equal partitioning of mitochondria between the two emerging daughter cells [19]. Mitotic division of mitochondria depends on the phosphorylation of Drp1 at Ser616 by cyclin B-cyclin-dependent kinase 1 (Cdk1) kinase.

The ability of the mitochondrial network to dynamically change its interconnected morphology to a fragmented one is key to the initiation of stress signaling pathways that trigger mitophagy or apoptotic responses. However, the molecular mechanisms underpinning these processes are largely unknown [10,20]. Several other proteins have been found to regulate mitochondrial dynamics (Table 1).

### 2.2. The Crosstalk between Mitochondrial Shape Changes and Reactive Oxygen Species

Several key components of the mitochondrial dynamics machinery have been demonstrated to depend on redox cues for activation in various disease settings [3]. For example, previous work has shown that β-amyloid nitrosylates Drp1 at Cys644, which leads to its activation, subsequent mitochondrial fragmentation, and neuronal injury suggesting that NO-regulated mitochondrial fission may be a key contributor to Alzheimer’s disease (AD) progression [57]. Drp1 was transnitrosylated via the intermediacy of Cdk5 in another AD model [35]. Similarly, Watanabe et al. have observed enhanced mitochondrial network fragmentation and mitochondrial function decline, which was associated with increased insulin resistance, following H_2_O_2_ treatment in H9c2 cardiomyoblasts and these effects occurred in a Drp1-dependent manner [58]. In addition, both overexpression of long-chain acyl-CoA synthetase 1 and long-term palmitate treatment has been correlated with increased levels of ROS and 4-hydroxy-2-nonenal, a marker of lipid peroxidation, in a cardiomyocyte lipotoxicity model [59]. These mitochondrial shape changes were attributed to reduced phosphorylation of Drp1 at Ser637 and altered proteolysis of Opa1. A thiol reductase activity was found to be imperative for normal mitochondrial homeostasis in endothelial cells since protein disulfide isomerase A1 (PDIA1) knockout mice displayed aberrant phenotypes such as ER-facilitated and Drp1-dependent mitochondrial fission and increased mitochondrial ROS generation resulting in cellular senescence [51]. The mechanism was shown to involve a specific PDA1-Drp1 interaction as PDIA1 deficiency led to Drp1 sulfenylation at Cys644. Nevertheless, it remains to be investigated further whether H_2_O_2_ activates Drp1 directly or indirectly. Lastly, ionizing radiation in the form of γ-rays or α particles induced mitochondrial fragmentation and accelerated mitochondrial superoxide formation in immortalized human fibroblasts or human small airway epithelial cells, which was abrogated by Drp1 knockdown or inhibition, respectively [60].

On the contrary, much less is known about the role of ROS in modulating pro-fusion proteins. In this respect, one of the pivotal finding was the observation that oxidized glutathione promotes mitofusin activation [61]. Cys684 of Mfn2 was later identified as the key thiol switch responsible for regulating mitochondrial fusion induced by oxidized glutathione. To outline, both mitochondrial fission and fusion proteins are subject either to direct or indirect redox regulation pathways.

### 2.3. Damaged Mitochondrial Fragments Are Cleared by Mitophagy

Organelle-specific types of autophagy act on discrete cellular compartments to enzymatically digest and recycle their biochemical cargo inside autophagosomes that fuse with lysosomes [62]. In contrast to general autophagy, which degrades cellular milieu non-specifically, mitophagy selectively removes dysfunctional and fragmented organellar segments. Upon mild stress, mitochondrial fragments compromised by deteriorating mitochondrial membrane potential are selectively pinched off the continuous mitochondrial network reticulum by the action of the mitochondrial fission machinery [63]. Subsequently, fragmented mitochondria are selectively recognized by the mitophagy machinery through receptor binding between OMM and the growing membranous structure termed the phagophore [64]. Following further extension, phagophore ultimately encapsulates the damaged mitochondrion into a new structure called the mitophagosome [65]. This vesicular organelle is composed of four membrane layers, two originate from the engulfed mitochondrion and two from the phagophore itself. In the final step, lysosomal fusion with mitophagosomes results in acidic hydrolysis of the mitochondrial content [65].

There are two basic modes by which the mitophagy assembly sequesters mitochondria for degradation depending on the involvement of ubiquitin and whether the interaction is direct or indirect [66]. In both cases, the interaction is mediated between microtubule-associated protein 1 light chain 3 (LC3), found on the phagophore, and the mitochondrial substrates [67]. In the ubiquitin-dependent pathway, LC3 indirectly binds to various polyubiquitinated OMM proteins, such as voltage-dependent anion channel (VDAC), Tom20, Mfn1, or Mfn2 [66,68]. These interactions are mediated by several factors including dual-domain adapter proteins such as p62 (also known as sequestosome 1), nuclear dot protein 52 (NDP52), Tax1 binding protein 1 (TAX1BP1), neighbor of BRCA1 gene 1 (NBR1), and optineurin. In this configuration, the adapter protein contacts LC3 through its LC3-interacting region (LIR) and binds to ubiquitin through its ubiquitin-binding domain (UBD) [69]. In the ubiquitin-independent pathway, LC3 directly interacts with specific OMM receptors such as Bcl-2/adenovirus E1B 19 kDa protein-interacting protein 3 (BNIP3) and NIX (also known as BNIP3L) [64].

One of the most intriguing mechanisms by which mitochondria are marked for lysosomal degradation via ubiquitin-dependent mitophagy is the PTEN-induced putative kinase 1 (PINK1)-Parkin system [66]. PINK1 is a serine/threonine kinase that is continuously imported into the matrix of energized mitochondria by the translocases of the outer (TOM) and inner (TIM) mitochondrial membrane [70]. During this process, PINK1 is proteolytically modified by the action of mitochondrial processing peptidase (MPP) and presenilin-associated rhomboid-like protease (PARL). The truncated form of PINK1 is consequently transported for proteasomal degradation into the cytosol. Dissipation of the mitochondrial membrane potential, which can be experimentally induced by the addition of a chemical uncoupler such as carbonyl cyanide *m*-chlorophenyl hydrazine (CCCP), hinders mitochondrial import of PINK1 and results in its stabilization at the OMM as an “eat-me” signal. Mitochondrial accumulation of PINK1 recruits the E3 ubiquitin ligase Parkin to the mitochondrial surface, where it polyubiquitinates mitochondrial substrates to tag the dysfunctional organelle for degradation in the lysosome.

### 2.4. The Complex Regulation of Intrinsic Apoptosis

There are two broad types of apoptotic cell responses, intrinsic and extrinsic. Whereas extrinsic apoptosis relies on external stimuli for activation, which are sensed by plasma membrane death receptors, intrinsic apoptosis proceeds via a mitochondrial-dependent stress-response pathway [71]. Intrinsic apoptosis can be triggered by both exogenous or endogenous factors such as oxidative stress, DNA damage, mitochondrial membrane potential collapse, and growth factor deprivation, or by developmental and physiological cues. For example, oxidative stress is sensed by distinct redox-signaling pathways that regulate activity of the mitochondrial dynamics machinery as a common platform for governing cell survival or apoptosis [3]. In the first line of defense, cells adapt to mild or transient stress by triggering mitophagy (Figure 1). The fragmented state of mitochondria facilitates the progression of mitophagy as a quality control mechanism that selectively removes dysfunctional mitochondria [10]. If the stress becomes severe or is persistent, fragmented mitochondrial phenotype leads to the activation of intrinsic apoptosis [3].

Apoptosis is tightly regulated by members of the B-cell lymphoma 2 (Bcl-2) family. Depending on their characteristic content of highly conserved Bcl-2 homology (BH) domains (BH1–4), these proteins can be divided into either pro-apoptotic or pro-survival [72]. Under basal conditions, the three domain-containing (BH1–3) effectors of apoptosis Bax and Bak are mostly localized to the cytosol and OMM, respectively, in their compact non-oligomeric forms [73]. Notably, Bax is believed to be in constant equilibrium between the cytosol and the OMM [74]. Upon apoptotic stimuli, rates of translocation of Bax to OMM exceed those of its retrotranslocation back into the cytosol resulting in its increased mitochondrial retention [75]. This is accompanied by conformational changes in the tertiary structure of Bax and its insertion into the outer leaflet of the OMM via its C-terminal helix (α9) [76]. The progression of early stages of intrinsic apoptosis can be interrogated by means of cell biology and molecular techniques directed at detecting mitochondrial localization of Bcl-2 proteins and the membrane insertion or activation status of Bax or Bak. Pro-survival proteins such as Bcl-2, B-cell lymphoma-extra large (Bcl-xL), and myeloid cell leukemia 1 (Mcl-1), which contain all four BH domains (BH1–4), were implicated in promoting cytosolic Bax sequestration to prevent cell death [77,78]. Direct activation or derepression of negative regulation by pro-survival Bcl-2 proteins is mediated by the BH3-only pro-apoptotic sub-class of proteins. This activity leads to additional conformational changes in Bax and Bak structure that facilitate their oligomerization inducing outer membrane permeabilization (MOMP) and release of pro-apoptotic factors such as cytochrome c (cyt c), apoptosis-inducing factor (AIF), second mitochondrially-derived activator of caspase/direct inhibitor of apoptosis-binding protein with low pI (Smac/DIABLO), Omi/high temperature requirement protein A2 (Omi/HtrA2), adenylate kinase 2, deafness/dystonia peptide (DDP)/TIMM8a and endonuclease G (endoG) [74,76,79,80]. The release of these second messengers into the cytosol following MOMP is considered the point of no return during intrinsic apoptosis as OMM integrity is permanently lost. In addition to spatiotemporal tracking of these second messengers, dissipation of mitochondrial membrane potential may serve as an indirect measure of MOMP progression. Importantly, novel nanoscopic techniques provided a proof-of-concept evidence for MOMP by visualizing Bax-mediated pores following apoptotic stimulation at a resolution beyond the diffraction limit [81].

Release of cyt c from IMM into the cytosol is essential for the allosteric activation of the downstream cascade of aspartate-specific cysteine endoproteases termed caspases [82]. Initially, cyt c binds to apoptotic protease activating factor-1 (Apaf-1) in a dATP/ATP-dependent manner, which induces a conformational change in this adapter protein. Upon nucleotide release, Apaf-1-cyt c heterodimers oligomerize into a multi-subunit heptameric complex called the apoptosome. Mature apoptosome subsequently activates the initiator caspase-9 through its caspase recruitment domain (CARD) [83]. Caspase-9, in turn, binds and proteolytically activates executioner caspases 3 and 7. Furthermore, caspase activation can also be effected by a derepression mechanism, which involves relieved inhibition of caspases from the inhibitor of apoptosis (IAP) protein family members, such as IAP1, IAP2, or x-linked inhibitor of apoptosis (XIAP), by Smac/DIABLO or Omi/HtrA2 [84]. As part of the execution phase of regulated cell demise, these caspases dimerize and activate a multitude of proteolytic and nucleolytic processes leading to the breakdown of nuclear DNA (nDNA) and regulated decomposition of the cell [85]. The proteolytic signal is further amplified and propagated by autocatalytic activation among the executioner caspase enzymes themselves. Major targets of executioner caspases include poly(ADP-ribose) polymerase (PARP), cytokeratin 18 (CK18), and inhibitor of caspase-activated DNase (iCAD). Alternatively, apoptosis may proceed through a caspase-independent pathway, in which AIF and endoG cooperate to directly induce nDNA fragmentation and chromatin condensation.

### 2.5. Mitochondrial Fission Is the First Step in Intrinsic Apoptosis

Numerous reports have investigated the causal link between the mitochondrial morphology and apoptotic activation [86,87,88,89,90,91]. The classic observation is that inhibition or genetic inactivation of Drp1 delays cyt c release and cell death via apoptosis [92]. Moreover, Bax has been shown to be recruited to Drp1-containing mitochondrial scission sites in mammalian cell lines following apoptotic stimuli such as staurosporine treatment [93]. Although overexpression of a dominant negative mutant allele of Drp1 (dn-Drp1) had no effect on Bax translocation, dn-Drp1-expressing cells were more resistant to staurosporine-induced apoptosis. Using Drp1 knockdown, a follow-up work pinpointed the stimulatory effect of Drp1 on apoptosis to occur downstream of mitochondrial Bax association but prior to the onset of MOMP [94]. Collectively, this body of evidence suggests that mitochondrial fragmentation promotes apoptosis at the level of Bax activation at the mitochondria and that cell death can be delayed by inactivating mitochondrial fission. In another study, researchers have found that Mfn1- or Mfn2 mitofusin-deficient mouse embryonic fibroblasts (MEFs) display a fragmented mitochondrial phenotype and are more sensitive to cisplatin treatment [95]. At the molecular level, increased sensitivity to cisplatin positively correlated with Bax activation and its insertion into the OMM as well as with the release of cyt c into cytosol. Furthermore, overexpression of Mfn1 or Mfn2 prevented mitochondrial fragmentation, cyt c release, and apoptosis upon cisplatin or azide treatment in cervical cancer HeLa cells when compared to empty vector control [95]. Analogous observations were made following overexpression of dn-Drp1 and following azide treatment in rat renal proximal tubule cells. Consistently, there was no difference observed in the translocation of Bax to mitochondria between the empty vector control and either Mfn1, Mfn2, or dn-Drp1 overexpressing HeLa cells following azide treatment suggesting that mitochondrial dynamics modulates intrinsic apoptosis downstream of mitochondrial Bax recruitment. In line with these results, overexpression of Mfn2 prevented cell death in cerebellar granule neurons exposed to various stress stimuli [96]. Although under very specific and unique conditions, such as following Fis1 overexpression, mitochondrial fragmentation was shown to drive the progression of intrinsic apoptosis, the consensus view is that mitochondrial dynamics changes only sensitize cells to pro-apoptotic stimuli but do not necessarily compromise cell viability [97]. This notion is congruent with the reversible and flexible nature of the mitochondrial network ultrastructure. Contrastingly, other reports have described an inhibitory effect of mitochondrial fission on RCD suggesting a greater degree of molecular and regulatory complexity in the underlying processes. Altogether, these findings demonstrate that there is, indeed, an interplay between the mitochondrial fission and apoptotic machineries and that mitochondrial fission-induced apoptotic induction may be a conserved feature of both transformed and non-transformed cells. The mechanism by which mitochondrial architecture remodeling predisposes cells to apoptosis is still an ongoing scientific debate. Increased membrane curvature has been shown to both promote and inhibit the recruitment of Bcl-2 family members to synthetic membrane vesicles therefore giving inconclusive answers. In an elegant study aimed at resolving this conundrum, Montessuit et al. proposed that Bax oligomerization may predominantly proceed at the constriction sites of mitochondrial membrane hemi-fission or hemi-fusion intermediates and that cyt c release may therefore likely be concentrated to these microdomains [98].

How can apoptotic proteins be targeted for cancer therapy? The inherent resistance of tumors to intrinsic apoptosis can be attributed to dysregulation of mechanisms at each of the molecular steps of intrinsic apoptosis. In general, therapeutic approaches aim to restore the delicate balance between pro-apoptotic and anti-apoptotic proteins. Given that the increased expression levels of Bcl-2, Bcl-x_L_, and Mcl-1 are frequently observed in tumor tissues, these proteins constitute important therapeutic targets for cancer therapy [84]. On the other hand, forcing cells into apoptosis by activating the BH3-only class of proteins using small molecule compounds known as BH3 mimetics represents an alternative path to cancer therapy [99]. The fact that antiapoptotic members of the Bcl-2 family are frequently mutated in cancers adds an extra layer of complexity in targeting these proteins. In summary, mitochondrial fission plays a vital role in the initiation of the reversible phase of intrinsic apoptosis and manipulating the activity of this system can alter cell sensitivity to anti-cancer drugs.

## 3. Reactive Oxygen Species and Cancer—A Dangerous Liaison

### 3.1. The Good and the Bad of Mitochondrial Reactive Oxygen Species

Owing to the fact that mitochondria harbor Complex IV as the terminal electron acceptor for oxygen utilization, these organelles are the largest intracellular source of oxygen radicals [100]. ROS bear the harmful potential to oxidize lipids, nucleic acids, and proteins making mitochondria the etiological origin of metabolic and degenerative diseases as well as aging [101,102,103]. ROS are unavoidably generated by enzymes of the mitochondrial electron transport chain (ETC) during side reactions of electrons with molecular oxygen [104]. Other mitochondrial oxidoreductases reported to generate ROS are glycerol-3-phosphate dehydrogenase, dihydroorotate dehydrogenase, electron-transfer flavoprotein-ubiquinone oxidoreductase (ETF:QO), proline oxidase, and the dihydrolipoamide dehydrogenase subcomplex of α-ketoacid dehydrogenases including pyruvate, α-ketoglutarate, α-ketoadipate, and branched-chain α-ketoacid dehydrogenase [105]. Initially, superoxide anion radical (O_2_^●−^) is produced by one-electron reduction of oxygen, which occurs predominantly at Complex I and III redox sites [106]. Although superoxide has limited reactivity and does not cross phospholipid membranes, it can be readily converted to H_2_O_2_, a nonradical ROS, by superoxide dismutase (SOD) or non-enzymatically [107]. The family of SOD enzymes is comprised of three different isoforms depending on cellular localization. Whereas copper/zinc isoforms of superoxide dismutase, SOD1 and SOD3, reside within mitochondrial inter-membrane space (IMS) and extracellular space, respectively, the manganese superoxide dismutase SOD2 is localized within the mitochondrial matrix [108].

One of the most active and damaging radical species is the hydroxyl radical (HO^●^) [109], which can be produced from H_2_O_2_ in the presence of divalent metal ions, such as ferrous iron, by the Fenton reaction [110]:Fenton reaction: H_2_O_2_ + Fe^2+^ → HO^●^ + OH^−^ + Fe^3+^

The Fenton reaction can be extended into so-called Haber–Weiss reaction (see below, top reaction), in which the iron ion acts as a catalyst, while cycling between its reduced and oxidized state, for both the Fenton (bottom left) and superoxide oxidation (bottom right) reactions [111]:$$\mathrm{Haber}-\mathrm{Weiss}\;\mathrm{reaction}:\;H_{2}O_{2}\;+\;O_{2}{}^{●}{}^{-}\;\overset{\mathrm{Fe}}{\to }\;HO^{●}\;+\;OH^{-}\;+{\;O}_{2}$$



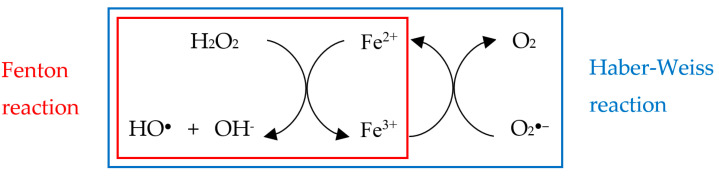



Other ROS types include singlet oxygen (^1^O_2_), nitric oxide (NO^●^), peroxynitrite (ONOO^−^), alkoxyl radical (RO^●^), peroxyl radical (ROO^●^), and hypochlorous acid (HOCl). Among these ROS, ^1^O_2_ has attracted eminent attention due to its central role as the active species in photodynamic therapy (PDT) [112], whereas NO^●^ and ONOO^−^, the only ROS that overlap with the family of reactive nitrogen species (RNS) [113]. NO^●^, produced by nitric oxide synthase in an oxygen-mediated NADPH-dependent oxidation of l-arginine to l-citrulline, is freely diffusible albeit poorly reactive toward biomolecules. Nonetheless, NO^●^ can spontaneously and rapidly react with superoxide to yield a highly promiscuous oxidant peroxynitrite:NO^●^ + O_2_^●−^ → ONOO^−^

The mode by which radicals induce mutagenic DNA lesions entails direct damage to nitrogen bases or DNA breaks. For example, hydroxyl radical-mediated oxidation of guanine yields 8-oxo-7,8-dihydroguanine, which tends to pair with adenine to give rise to a transversion mutation during replication. Given its close vicinity to ETC and the general sparsity of mtDNA repair mechanisms, the mutation rate of mtDNA is estimated to be around ten times higher than that of nDNA [114,115,116].

Oppositely to the harmful role originally proposed, the signaling potential of ROS has been realized soon after their initial discovery [117]. The half-life of H_2_O_2_ is relatively long and as a small, compact, and uncharged molecule, H_2_O_2_ is best suited, among all other ROS types, to serve a signaling function [118]. Indeed, H_2_O_2_ can freely permeate biological membranes, diffuse out of mitochondria, and equilibrate within various cellular compartments to effect physiological responses. In addition, ROS are known to modify amino acid side chains of proteins, particularly the sulfhydryl group of cysteine. In fact, cysteine represents an important relay switch that reversibly alternates between the oxidized and reduced state of redox-sensitive proteins, which are the integral part of redox-mediated signaling cascades, and that of the tripeptide glutathione (γ-Glu-Cys-Gly), which forms the basis for antioxidant redox buffering [3,119,120].

### 3.2. The Endogenous Antioxidant Defense Machinery

Intracellular and intramitochondrial levels of ROS are modulated by intricate antioxidant defense systems that scavenge superoxide, H_2_O_2_, as well as other oxygen radicals to protect cells from harmful effects of oxidative stress but also to fine-tune the redox microenvironment for optimal redox signaling [121]. The most abundant intracellular antioxidant molecule is glutathione, which cycles between its reduced (GSH) and oxidized (GSSG) forms [120]. GSH can be synthetized de novo from L-glutamate, cysteine, and glycine by glutamate-cysteine ligase and glutathione synthetase with γ-glutamylcysteine as an intermediate. Among the ROS detoxifying enzymes, superoxide dismutase (SOD) and catalase play the most prominent role as their concerted activity allows for the net reduction of superoxide to H_2_O. SOD reaction involves the disproportionation of two superoxide molecules to oxygen and hydrogen peroxide:Superoxide dismutase: 2O_2_^●−^ + 2H^+^ → O_2_ + H_2_O_2_

Owing to the role of mitochondrially-derived H_2_O_2_ as a key redox second messenger, the cytosolic enzyme catalase plays an important function in redox signaling pathways by decomposing H_2_O_2_ to oxygen and H_2_O:Catalase: 2H_2_O_2_ → O_2_ + 2H_2_O

H_2_O_2_ can be also neutralized to H_2_O by the NADPH-dependent glutathione peroxidase (Gpx) and glutathione reductase (GR) system:



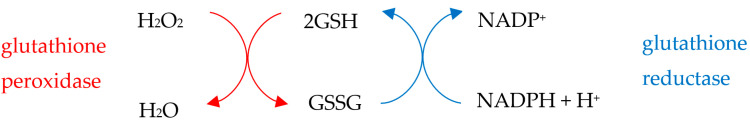



Reduction of oxidized proteins can be catalyzed by the thioredoxin (Trx) and glutaredoxin (Grx) relay mechanisms. The thioredoxin system, which consists of thioredoxin and thioredoxin reductase (TrxR), catalyzes the reduction of cysteine disulfide bridges through the active site motif Cys-Gly-Pro-Cys present on Trx:



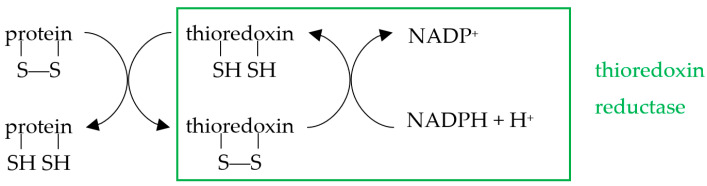



A common substrate for thioredoxin is peroxiredoxin, which elicits an antioxidant role by converting H_2_O_2_ to H_2_O:



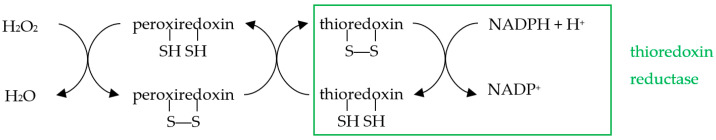



Correspondingly, the Grx system is composed of glutaredoxin and glutaredoxin reductase (GrxR):



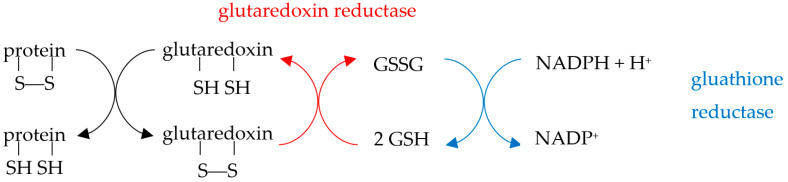



Detoxification mechanisms of the less stable radical species are relatively understudied, owing to technical difficulties associated with their detection in biological systems. Endogenous antioxidants capable of scavenging HO^●^ include melatonin, aromatic amino acid side chains, and disulfide bond-rich proteins such as fibrinogen or human serum albumin, although cyto-protection against HO^●^ is orchestrated also at the level of iron metabolism [122,123,124]. Moreover, dietary supplements such as vitamin C and E, β-carotene, as well as flavonoids are known quenchers of ^1^O_2_, potentially rendering PDT approaches less efficient [125,126]. Finally, vitamin C and E as well as melatonin were reported to scavenge RO^●^ and ROO^●^ radicals while uric acid efficiently neutralized ONOO^−^ [127,128,129]. To conclude, oxidative stress can be attributed to the imbalance in redox homeostasis.

### 3.3. Apoptosis Is a Redox-Dependent Process

Since apoptotic resistance is associated with tumor initiation, cancer progression, and metastasis, it is imperative to identify molecular processes that inactivate the safeguarding role of RCD [130]. Among the key factors contributing to apoptotic regulation is cellular redox homeostasis [131]. ROS were reported to both positively and negatively regulate apoptosis in a cell context-dependent manner [132]. For example, H_2_O_2_ activates and recruits Bax to mitochondria via a mechanism involving cytosolic acidification in colorectal carcinoma HCT116 cells [133]. This is consistent with the finding that Cys62 is required for the H_2_O_2_-mediated activation of Bax in SW480 human colon adenocarcinoma cells [134]. Similarly, Cys62 and Cys126 Bax residues were identified as intracellular redox sensors for the initiation of Bax conformational changes during apoptosis induced by selenite in in colorectal cancer cells. Using the general antioxidant *N*-acetyl-l-cysteine (NAC), Zheng et al. [135] have demonstrated that both Bax translocation and activation following As_2_O_3_ treatment in hematopoietic IM-9 cells occurred in a manner dependent on ROS. However, the redox sensing mechanism for directing mitochondrial Bax translocation remains unclear. Furthermore, the pro-oxidant state of Bax was necessary for cyt c release and apoptosis progression induced by nerve growth factor (NGF) withdrawal in mouse sympathetic neurons [136,137]. Taken together, these studies suggest that apoptosis may be regulated through the oxidative status of Bax.

Given that enhanced ROS generation may stem from both metabolically active and dysfunctional mitochondria, it may be instrumental to tease out which of these two modalities serves as the trigger for the intrinsic apoptotic form of RCD. Previous work has highlighted the fact that OXPHOS facilitates stress-induced Bax- and Bak-dependent apoptosis in human breast cancer MCF-7 and hepatoma HepG2 cell lines [138]. These observations are consistent with the notion that ROS mediate the link between mitochondrial respiration and apoptosis activation. Importantly, it has been suggested that it is the dependence of tumors on glycolytic metabolism—a phenomenon termed as the Warburg’s effect—that renders cancer cells resistant to pro-apoptotic insults. This would suggest that mitochondria are highly functional before cells commit to apoptosis and that metabolic reprogramming away from OXPHOS underlies cancer resistance. Mechanistically, the Warburg effect produces reducing equivalents that help neutralize oxidative stress generated in tumor cells due to elevated metabolism. The crosstalk between OXPHOS and apoptosis may be conserved across eukaryotes as it has been also described in yeast [139]. In contrast, inhibiting ETC activity may stimulate apoptosis through mitochondrial ROS production. Mechanistically, the site of ROS formation in human breast cancer tumors might be Complex III as withaferin A, a steroidal lactone from the withanolide family isolated from the plant Withania somnifera, stimulated apoptosis through Complex III inhibition while the effect was abolished upon SOD1 overexpression [140]. Conversely, ectopic expression of the single-subunit yeast NADH dehydrogenase Ndi1 in Drosophila melanogaster led to decreased mitochondrial ROS production, observed in isolated mitochondria respiring on Complex I substrates, along with reduced apoptosis revealed by TUNEL staining in brain sections of aged female flies [141]. To summarize, ROS-induced apoptosis can proceed through inducing mitochondrial function or dysfunction in a context-dependent manner.

### 3.4. Reactive Oxygen Species as a Double-Edged Sword for Cancer

Increased levels of ROS as a result of metabolic rewiring are characteristic for many tumor types as they are thought to promote proliferation and cancer cell survival during tumorigenesis [8]. Examples of tumors that were confirmed to have high rates of ROS formation include melanoma, neuroblastoma, liver hepatoma, ovarian, colon, pancreatic, and breast carcinoma [142,143,144,145,146]. Redox signaling plays a cardinal role during all stages of carcinogenesis with the source of ROS being both mitochondrial and cytosolic in origin [142,147]. Indeed, ROS has been reported to drive tumor initiation, transformation, proliferation, and dissemination [148]. Physiologically, elevated levels of ROS promote mutagenesis, proto-oncogene activation, tumor suppressor inactivation, stimulation of metabolic, autophagic, and signal transduction pathways, cell proliferation, survival, adaptation to unfavorable microenvironment such as that found within the core of solid tumor [149]. This tumor phenotype may be associated with resilience to therapeutic approaches, increased risk of developing metastasis, and poor patient prognosis.

Conversely, excessive accumulation of oxidative stress in some cancer types can lead to cell cycle arrest, apoptosis, or senescence [150]. Owing to its role in driving tumorigenesis, intracellular ROS can be looked upon not only as a marker for oncogenic transformation but also as a specificity guide for targeted anti-cancer therapy [151]. Given the differential rates of ROS generation found within a tumor, identifying therapeutic windows for pro-oxidant cancer therapy aimed to increase ROS levels above a putative threshold may be a viable paradigm toward eliminating predominantly cancer but not normal cells (Figure 2). Along these lines, endogenous ROS detoxifying systems, which are frequently upregulated in tumors to escape cytoprotective mechanisms, are also widely recognized as potential targets for anti-cancer therapy. In conclusion, ROS are a valid therapeutic target for cancer.

## 4. Surpassing the Redox Threshold for Apoptosis in Tumors by Inducing Mitochondrial Fission

### 4.1. Cyclin C Acts as a Bridge between Stress-Induced Mitochondrial Fission and Apoptosis

Activating mitochondrial fission, instead of inhibiting it, to combat cancer is a relatively novel concept [87,152]. Many of the previous chemotherapeutic attempts were focused on targeting pro-fission factors such as using the Drp1 inhibitor mdivi-1 [153]. These approaches were predominantly motivated by upregulation of Drp1 and the consequent fragmented mitochondrial phenotype morphology commonly observed in cancers. As an alternative, one can imagine a strategy whereby a pulse of mild oxidative stress would activate the mitochondrial fission-apoptosis redox axis (Figure 3).

Indeed, such a model has been verified experimentally. Pretreatment of the model cervical cancer HeLa cells with a stapled peptide mimetic (S-HAD) that disrupts the interaction between the highly conserved tumor suppressor cyclin C, a transcriptional regulator canonically found in the nucleus, and the Mediator, a transcriptional coactivator complex of RNA polymerase II, led to effective sensitization of these cells to cisplatin treatment [87]. The rationale behind this strategy was based on previous experiments in yeast wherein cyclin C is released from Med13p binding in the nucleus following various forms of cellular stress. Cytoplasmic cyclin C induced extensive fragmentation of the mitochondrial network [32]. Moreover, depletion of Med13p was sufficient to induce mitochondrial fission and sensitized yeast cells to H_2_O_2_ treatment [154]. In fact, Med13p destruction was later shown to be a critical part of the oxidative stress sensing mechanism that regulates nuclear release of cyclin C, which relies upon the mitogen-activated protein kinase (MAPK) signaling pathway for activation [155]. Summarily, nuclear cyclin C release represents a bona fide redox signaling pathway impinging on mitochondrial dynamics proteins.

What is the mechanism by which cytosolic cyclin C effects mitochondrial dynamics and cell viability? The role appears to be direct and independent of the transcription regulatory function of cyclin C. In mammalian and yeast cell cultures, cyclin C was shown to interact with Drp1 and Dnm1, the yeast ortholog of Drp1, respectively, to promote mitochondrial fission and stress-induced apoptosis [31,32]. The association between cyclin C and Drp1 was also analyzed in vitro [17]. The results indicated that cyclin C binds the GTPase domain of Drp1 through its second cyclin box domain. Nevertheless, the most fundamental observation that pointed to the pro-apoptotic function of cyclin C as a solid tumor suppressor was that MEF cells lacking cyclin C were resistant to cisplatin-induced apoptosis [31]. Unraveling the mechanism of the stress response pathway mediated by cyclin C further revealed that S-HAD or H_2_O_2_ treatment led not only to mitochondrial fragmentation but also to enhanced mitochondrial Bax recruitment and activation and this occurred in both cyclin C- and mitochondrial ROS-dependent manner [87]. Curiously, although S-HAD-induced mitochondrial fragmentation positively correlated with increased production of mitochondrial superoxide, S-HAD treatment had no significant effect on cell viability. Hence, it is the contribution from three oxidative stresses, one from the transformed tumor itself, second from cisplatin, and the third from S-HAD-induced mitochondrial fission, that act in synergy to surpass the threshold for apoptotic cancer cell killing. Accordingly, one can conjecture that it is not the fragmented phenotype per se that leads to the activation of apoptotic processes but instead the sudden change in mitochondrial morphology, which is required to trigger the maximum production of ROS.

Further evidence for a role of cyclin C as a tumor suppressor was demonstrated in organs isolated from thyroid-specific [156] and pancreas-specific (R.S. and Kerry S. Campbell, Fox Chase Cancer Center, Philadelphia, PA unpublished observations) *CCNC* knockout mouse. Despite the mitochondrial fission-promoting response of cyclin C to H_2_O_2_ or S-HAD could be recapitulated in murine poorly differentiated thyroid cancer (PDTC) cell line, this effect may be uncoupled from Bax activation as there was no difference observed between the numbers of apoptotic cells in the control and *CCNC* siRNA cells following cisplatin treatment. Overall, these data suggest that pharmacological activation of the cyclin C pathway may be a promising chemotherapeutic strategy to treat solid tumors. However, more investigations are needed to confirm the utility of such a combination chemotherapy approach, particularly with respect to taking advantage of a xenograft mouse model. A careful examination of the mitochondrial fission process will also be needed to elucidate the spatiotemporal kinetics of mitochondrial function before the onset of MOMP. Namely, the principal question to be addressed in the future therefore is whether mitochondrial fission-stimulated ROS production is a result of increased or decreased mitochondrial function? Also, pinpointing the source of the mitochondrial ROS would help to contribute to a more thorough understanding of the underlying signaling mechanism.

### 4.2. Does a Self-Perpetuating Cycle of Mitochondrial Fission and Reactive Oxygen Species Production Underpin the Mechanism of Apoptotic Sensitization?

Notwithstanding the exact mechanism involved, one may speculate whether there is a self-potentiating continuum between redox sensing, such as mediated by cyclin C, and mitochondrial fission-induced ROS production, which may contribute to more effective killing of cancer cells upon the administration of chemotherapy co-adjuvants stimulating mitochondrial fission, such as S-HAD (Figure 4). This scenario assumes that, in addition to the nuclear-to-mitochondrial signaling effected by cyclin C, there is a retrograde communication between mitochondria and the nucleus mediated by an unknown second messenger, which could potentially be H_2_O_2_. Regarding the molecular mechanism involved, we can only speculate whether the retrograde signaling role is solely elicited by H_2_O_2_ or whether additional ROS-dependent processes such as mitophagy or lipid peroxidation are being concurrently activated. Although the proposed signaling mechanism inarguably deserves future scrutiny, the resultant net effect of S-HAD treatment is the sensitization of tumors to apoptosis-inducing chemotherapy [87].

## 5. Conclusions

Although the role of redox equilibria in cancer etiology is far from being clear cut, several generalizing principles can be extracted. First, tumors are more likely to have a higher content of reactive oxygen radicals, a phenomenon that can be exploited for pro-oxidant cancer therapy. Second, mitochondrial dynamics may play an integral role in oncogenesis. Third, activating the acute capacity of fragmented mitochondria to generate ROS may contribute to targeted cancer cell killing via the apoptotic pathway.

## Figures and Tables

**Figure 1 biology-10-00033-f001:**
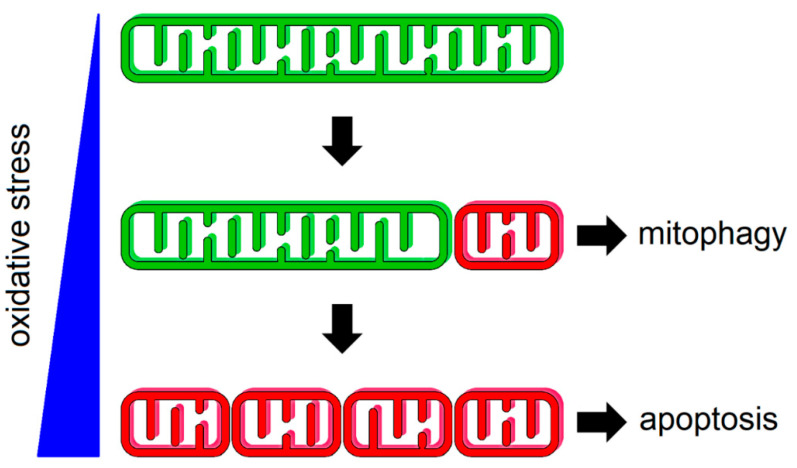
Mitochondrial network responds to oxidative stress by fragmentation. Mild oxidative stress leads to the scission of individual dysfunctional mitochondrion (red) from functional organelles (green), which are cleared by mitophagy. Persistent oxidative stress results in more profound mitochondrial fragmentation, thus predisposing cells to death through apoptosis. A large variety of chemotherapeutic agents elicit their cytotoxicity through elevating the oxidative status of the cell.

**Figure 2 biology-10-00033-f002:**
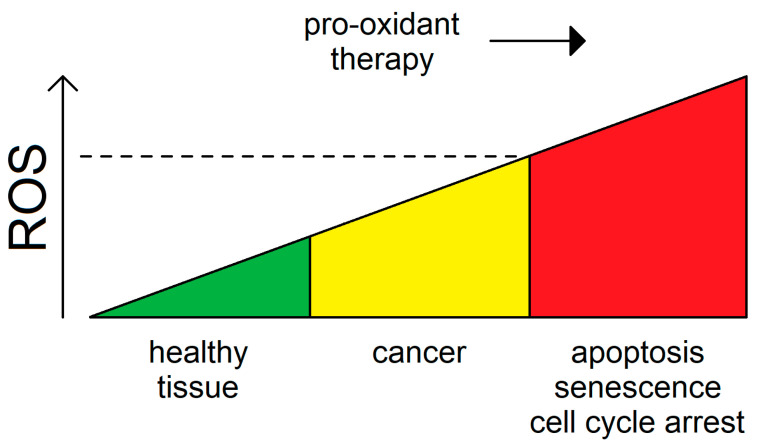
The role of ROS in tumorigenesis. Cancer cells tend to upregulate ROS to support their metabolic and proliferative potential. Increased oxidative stress may, in turn, contribute to the mutational burden of these tumors. Pro-oxidant chemotherapeutic or radiotherapeutic approaches may be deployed to elicit cell cycle inhibitory, apoptotic, or senescent outcomes after overcoming a specific threshold, which is distinctive between cancer and the healthy tissues (dashed line).

**Figure 3 biology-10-00033-f003:**
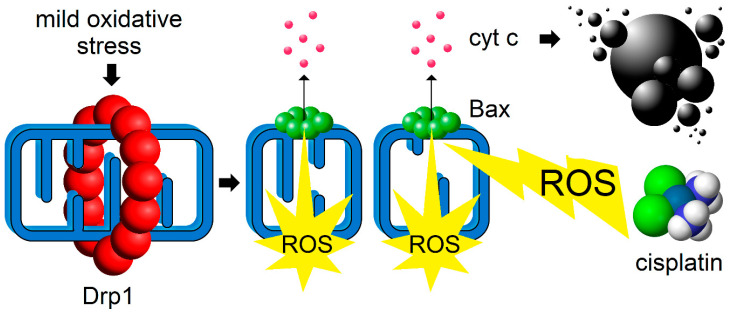
Inducing mitochondrial fission process as a co-adjuvant approach to current chemotherapeutic strategies. Pharmacological interventions aimed at inducing mild oxidative stress can lead to Drp1 activation (red), increased mitochondrial fragmentation and ROS production. This, in turn, activates pro-apoptotic effector protein Bax (green), which oligomerizes to induce MOMP-dependent cytochrome c (cyt c) release (pink) thereby rendering tumor cells (black) more sensitive to ROS-inducing anti-cancer drug such as cisplatin.

**Figure 4 biology-10-00033-f004:**
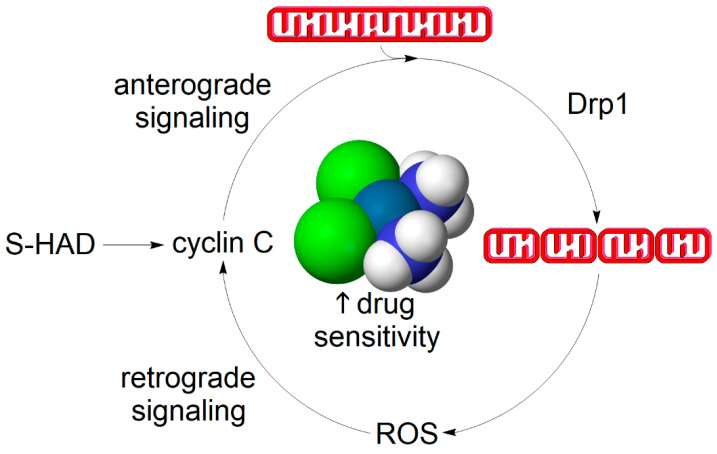
A hypothetical self-propelling cycle between mitochondrial fission and ROS generation leading to the mitochondrial fragmentation phenotype. Cyclin C association with its nuclear anchor Med13 is disrupted by treatment with the stapled peptide mimetic of the N-terminal domain of cyclin C (S-HAD) or upon exogenous oxidative stress such as H_2_O_2_ or cisplatin treatment. Cyclin C is partially released from the nucleus to bind and activate Drp1 and thereby induce mitochondrial fission. Mitochondrial fragments are more likely to emanate ROS signals such as in the form of H_2_O_2_ derived from mitochondrially-produced superoxide. This could be either due to mitochondrial dysfunction or a transient rise in respiratory function. Nevertheless, mitochondrial ROS-mediated retrograde signaling may act on cyclin C through an unknown redox sensor. The closing of the positive feedback loop between mitochondrial fission and mitochondrial ROS formation contributes to rapid amplification of the redox signal, also referred to as ROS burst. Advantageously, tipping ROS levels over apoptosis-inducing threshold through generating an ROS burst, such as using S-HAD, renders cancer cells more sensitive to chemotherapeutic interventions [87]. Other signaling pathways may participate in relaying the redox signal to the mitochondrial fission machinery as well and mitochondrially-derived ROS may short-circuit this cycle by directly modulating Drp1 activity [3].

**Table 1 biology-10-00033-t001:** Regulators of mitochondrial dynamics.

Name.	Abbreviation	References
Abelson tyrosine kinase	c-Abl	[21]
AMP-activated protein kinase	AMPK	[22,23]
calcineurin	–	[24,25,26,27]
Ca^2+^/calmodulin-dependent protein kinase	CaMK	[28,29]
c-Jun N-terminal kinase	JNK	[30]
cyclin C	–	[31,32]
cell division cycle 20 related 1	Cdh1	[33]
cyclin-dependent kinase 1	Cdk1	[19,24]
cyclin-dependent kinase 5	Cdk5	[34,35]
cyclophilin D	–	[36]
death-associated protein 3	DAP3	[37]
endophilin B1	–	[38]
extracellular signal-regulated kinase	Erk	[39]
ganglioside-induced differentiation associated protein 1	GDAP1	[40]
hypoxia-inducible factor 1α	HIF1α	[41]
mitochondrial fission process 1	MTFP1	[42]
nuclear factor κB	NF-κB	[43]
nuclear factor (erythroid-derived 2)-like 2	Nrf2	[44]
overlapping activity with m-AAA protease 1	Oma1	[45]
mitofilin	–	[46]
p38 mitogen-activated protein kinase	p38 MAPK	[47]
PGAM5	–	[48]
phospholipase D	PLD	[49,50]
protein disulfide isomerase A1	PDIA1	[51]
protein kinase A	PKA	[27,52]
protein kinase C, isoform δ	PKCδ	[24]
Rho-associated coiled-coil containing protein kinase 1	ROCK1	[53]
S6 kinase 1	S6K1	[54]
uncoupling protein 2	UCP2	[55,56]

## Data Availability

Not applicable.

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
