# Peer review of "The Impact of Mitochondrial Fission-Stimulated ROS Production on Pro-Apoptotic Chemotherapy"

_biology, 2021, doi:10.3390/biology10010033_

Round 1
Reviewer 1 Report
A brief summary
In this review, the authors discuss the regulation of mitochondrial fission-fusion, the role of mitochondrial fission in promoting apoptosis, and how this can be targeted as an anticancer therapy. More specifically, the authors propose, based on the current literature, that a H2O2-dependent increase in mitochondrial fission can cause a further mitochondrial derived H2O2 increase, which ultimately leads to a H2O2-dependent induction of apoptosis. This process seems to be very interesting to exploit as an anticancer therapy, because cancer cells are often characterized by overall higher H2O2 levels compared to untransformed cells.
Broad comments
The authors discuss an interesting and (as the authors mention) relatively new model for cancer therapy. In general, the authors provide a clear and concise overview of the relevant literature. They also present the different topics in such a manner that it builds up well to the model they propose, making this a relevant and hypothesis generating synthesis of the current literature in this field.
I have the following broad and specific comments that I think could help improve the manuscript even further.
- The authors discuss the role of superoxide/H2O2 as signaling molecule and the role of ROS as uncontrolled damaging agents in disease. They further discuss the H2O2-dependent regulation of cyclin C mediated mitochondrial fission and H2O2-dependent regulation of apoptosis, for example via targeted oxidation of Bax. In their final model, they propose a positive feedback loop between H2O2-dependent mitochondrial fission and mitochondrial H2O2 production, ultimately causing H2O2-dependent apoptosis. While it appears this model falls under the umbrella of H2O2 as signaling molecule, rather than ROS as uncontrolled damaging agent, the authors do not mention in their final model what the direct effects of H2O2/ROS could It would help future research aimed at addressing this model if the authors can incorporate in the different steps of their model what they think the role of H2O2/ROS is.
- The authors discuss 1) the mitochondrial fusion/fission machinery and 2) the H2O2-dependent regulation of nuclear cyclin C release, which then leads to mitochondrial fission. However, H2O2 also directly affects multiple components of the mitochondrial fusion/fission machinery, not only through nuclear cyclin C release. This seems highly relevant for their model, namely how would increased mitochondrial H2O2 release affect the fission/fusion machinery besides cyclin C? I therefore recommend incorporating a discussion of the H2O2-dependent regulation of the fission-fusion machinery, similar to “3.3. Apoptosis is a Redox-Dependent Process”, and also incorporate the information from this section in their model.
Specific comments
- line 20: “… mitochondrial fragmented phenotype and mitochondrial oxidative metabolism are interconnected within a self-propagating cycle”: Do the authors mean increased mitochondrial ROS production rather than mitochondrial oxidative metabolism?
- line 57-58: “On one hand, cancer cells produce excessive amounts of ROS to sustain increased metabolic needs for tumor growth [8].” Do the authors mean, excessive amounts of ROS as a byproduct of their changed metabolism to sustain increased metabolic needs for tumor growth?
- Video S1: CellMask Orange is used to delineate the plasma membrane. However, the cell plasma membrane is not clearly visible in these cells, and instead an organellar structure within the cytoplasm that is different from mitochondria is being stained. The authors need to explain what is being stained by CellMask Orange, or rather leave out the CellMask Orange signal altogether. Secondly, it would be very informative if some examples of fission and fusion that occur within the video could be highlighted.
- Line 129: Do the authors mean” …by the action of …”
- Line 144-146: What do the authors mean with this sentence, when talking about intrinsic apoptosis regulation: “For example, oxidative stress is sensed by distinct redox signaling pathways that regulate activity of the mitochondrial dynamics machinery as a common platform for producing cell fate decisions [3].” Do they mean “.. as a common platform for cell survival or apoptosis”? I recommend making this sentence clearer.
- Line 146: What is referred to with ”In the first instance”?
- What is the relevance of discussing markers for the detection of apoptosis in line 203-215?
- Line219-220: The authors state “Numerous reports have investigated the causal link between the mitochondrial ultrastructure and apoptotic activation [76].”, but only show one reference, which, as far as I could tell, did not look at mitochondrial ultrastructure directly. Further, do the authors mean mitochondrial morphology instead of ultrastructure here? Related to this, at line 28 and line 479: “Mitochondrial ultrastructure” is used, but do the authors mean “mitochondrial morphology”?
- The section 3.1 discusses the signaling and damaging roles of ROS. However, the two roles seem to be discussed somewhat randomly, which will be confusing for non-experts. For example, line 311-315 mentions the effect of ROS & guanine oxidation on introducing DNA mutations, and is immediately followed by the effect of ROS& cysteine oxidation in proteins, which is a key aspect of physiological redox signaling. I would recommend reorganizing this section to better distinguish between the role of superoxide/hydrogen peroxide in redox signaling and ROS in cell damage.
- Line 322-323: It is stated “…antioxidant defense systems that scavenge superoxide, H2O2, as well as other oxygen radicals…”. However, in the subsequent part the authors only mention antioxidant proteins that scavenge O2.- and H2O2, and not for example hydroxyl radical or singlet oxygen. To be complete, the authors need to mention which (if any) cellular protective mechanisms exist against these latter ROS.
- Line 455-458: “Moreover, depletion of Med13p was sufficient to induce mitochondrial fission and sensitized yeast cells to H2O2 treatment [115], suggesting that nuclear cyclin C release represents a bona fide redox signaling pathway impinging on mitochondrial dynamics proteins [4].” The reasoning in this sentence seems incorrect, since the first part says that release of cyclin C makes cells sensitive to H2O2, whereas the second part (and reference 4) says that H2O2 induces release of nuclear cyclin C. Therefore, the use of “suggesting” seems incorrect.
Author Response
Broad comments
- The authors discuss the role of superoxide/H2O2 as signaling molecule and the role of ROS as uncontrolled damaging agents in disease. They further discuss the H2O2-dependent regulation of cyclin C mediated mitochondrial fission and H2O2-dependent regulation of apoptosis, for example via targeted oxidation of Bax. In their final model, they propose a positive feedback loop between H2O2-dependent mitochondrial fission and mitochondrial H2O2 production, ultimately causing H2O2-dependent apoptosis. While it appears this model falls under the umbrella of H2O2 as signaling molecule, rather than ROS as uncontrolled damaging agent, the authors do not mention in their final model what the direct effects of H2O2/ROS could It would help future research aimed at addressing this model if the authors can incorporate in the different steps of their model what they think the role of H2O2/ROS is.
As of now, a detailed insight into the underlying redox mechanism is currently sought after, hence we included at the end of Section 4.2: “Regarding the molecular mechanism involved, we can only speculate whether the retrograde signaling role is solely elicited by H2O2 or whether additional ROS-dependent processes such as mitophagy or lipid peroxidation are being concurrently activated. Although the proposed signaling mechanism inarguably deserves future scrutiny, the resultant net effect of S-HAD treatment is the sensitization of tumors to apoptosis-inducing chemotherapy [Jezek, J.; Chang, K.T.; Joshi, A.M.; Strich, R. Mitochondrial translocation of cyclin C stimulates intrinsic apoptosis through Bax recruitment. EMBO Rep 2019, 20, e47425]” (lines 553–558).
- The authors discuss 1) the mitochondrial fusion/fission machinery and 2) the H2O2-dependent regulation of nuclear cyclin C release, which then leads to mitochondrial fission. However, H2O2 also directly affects multiple components of the mitochondrial fusion/fission machinery, not only through nuclear cyclin C release. This seems highly relevant for their model, namely how would increased mitochondrial H2O2 release affect the fission/fusion machinery besides cyclin C? I therefore recommend incorporating a discussion of the H2O2-dependent regulation of the fission-fusion machinery, similar to “3.3. Apoptosis is a Redox-Dependent Process”, and also incorporate the information from this section in their model.
A new section (2.2 The Crosstalk between Mitochondrial Shape Changes and Reactive Oxygen Species) has been written up (97–124) and a comment has been added to Figure 4 (old Figure 3) to emphasize this possibility in the proposed redox signaling model (576).
Specific comments
- line 20: “… mitochondrial fragmented phenotype and mitochondrial oxidative metabolism are interconnected within a self-propagating cycle”: Do the authors mean increased mitochondrial ROS production rather than mitochondrial oxidative metabolism?
Yes, the text has been edited accordingly (20).
- line 57-58: “On one hand, cancer cells produce excessive amounts of ROS to sustain increased metabolic needs for tumor growth [8].” Do the authors mean, excessive amounts of ROS as a byproduct of their changed metabolism to sustain increased metabolic needs for tumor growth?
Yes, the sentence has been edited (56).
- Video S1: CellMask Orange is used to delineate the plasma membrane. However, the cell plasma membrane is not clearly visible in these cells, and instead an organellar structure within the cytoplasm that is different from mitochondria is being stained. The authors need to explain what is being stained by CellMask Orange, or rather leave out the CellMask Orange signal altogether. Secondly, it would be very informative if some examples of fission and fusion that occur within the video could be highlighted.
The HepG2 cells were grown in 10 mM galactose in this experiment, which may explain the given phenotype. We made this fact clear in the legend to Video S1 (587) and appended the following comment: “The fact that CellMask Orange staining did not delineate the plasma membrane uniformly and led to the appearance of endosome-like vesicles may stem from the altered carbon metabolism (glucose replaced by galactose) utilized to force these cells to rely on OXPHOS for energy production [Jezek, J.; Plecita-Hlavata, L.; Jezek, P. Aglycemic HepG2 Cells Switch From Aminotransferase Glutaminolytic Pathway of Pyruvate Utilization to Complete Krebs Cycle at Hypoxia. Front Endocrinol (Lausanne) 2018, 9, 637]” (588–592). In addition, the video has been annotated for several examples of mitochondrial fission and fusion events.
- Line 129: Do the authors mean” …by the action of …”
Yes, the text has been edited as such (131).
- Line 144-146: What do the authors mean with this sentence, when talking about intrinsic apoptosis regulation: “For example, oxidative stress is sensed by distinct redox signaling pathways that regulate activity of the mitochondrial dynamics machinery as a common platform for producing cell fate decisions [3].” Do they mean “.. as a common platform for cell survival or apoptosis”? I recommend making this sentence clearer.
Yes, the text has been changed to “platform for governing cell survival or apoptosis” (173).
- Line 146: What is referred to with ”In the first instance”?
“Instance” has been changed to “line of defense” for better clarity (173).
- What is the relevance of discussing markers for the detection of apoptosis in line 203-215?
This section has been deleted.
- Line219-220: The authors state “Numerous reports have investigated the causal link between the mitochondrial ultrastructure and apoptotic activation [76].”, but only show one reference, which, as far as I could tell, did not look at mitochondrial ultrastructure directly. Further, do the authors mean mitochondrial morphology instead of ultrastructure here? Related to this, at line 28 and line 479: “Mitochondrial ultrastructure” is used, but do the authors mean “mitochondrial morphology”?
Several other relevant citations were included in this sentence. Furthermore, the term “mitochondrial ultrastructure” has been replaced with “mitochondrial morphology” (28, 232, 527).
- The section 3.1 discusses the signaling and damaging roles of ROS. However, the two roles seem to be discussed somewhat randomly, which will be confusing for non-experts. For example, line 311-315 mentions the effect of ROS & guanine oxidation on introducing DNA mutations, and is immediately followed by the effect of ROS& cysteine oxidation in proteins, which is a key aspect of physiological redox signaling. I would recommend reorganizing this section to better distinguish between the role of superoxide/hydrogen peroxide in redox signaling and ROS in cell damage.
Section 3.1 has been restructured to segregate the signaling and damaging aspects of ROS (302–344).
- Line 322-323: It is stated “…antioxidant defense systems that scavenge superoxide, H2O2, as well as other oxygen radicals…”. However, in the subsequent part the authors only mention antioxidant proteins that scavenge O2.- and H2O2, and not for example hydroxyl radical or singlet oxygen. To be complete, the authors need to mention which (if any) cellular protective mechanisms exist against these latter ROS.
Section 3.2 has been extended to cover several examples of endogenous scavengers of HO●, 1O2, RO●, ROO● and ONOO- (399–408).
- Line 455-458: “Moreover, depletion of Med13p was sufficient to induce mitochondrial fission and sensitized yeast cells to H2O2 treatment [115], suggesting that nuclear cyclin C release represents a bona fide redox signaling pathway impinging on mitochondrial dynamics proteins [4].” The reasoning in this sentence seems incorrect, since the first part says that release of cyclin C makes cells sensitive to H2O2, whereas the second part (and reference 4) says that H2O2 induces release of nuclear cyclin C. Therefore, the use of “suggesting” seems incorrect.
These statements were split into two independent sentences (503, 507) and incorporated into the text.
Reviewer 2 Report
I think the authors prepared very nice review on a current topic, relevant to a new type of potential cancer therapy, i.e., targeting mitochondria and ROS production. The reviewed area of research is important also to understand the redox biology of normal cells and how they can transform to cancer cells; the roles of mitochondria in this process can be predicted but remains only poorly understood. The article is very well prepared so I have only few possible suggestions how to possibly improve it.
What I miss i a schematic diagram showing mutual (and other important interactions) between proteins discussed in the text and involved in mitochondria "management", redox homeostasis and apoptosis. It would dramatically improve readability of the text.
If the manuscript becomes too long, the chapter about the apoptosis (that mostly summarizes the "textbook" knowledge) may be reduced.
As mitochondria are not only targeted by chemotherapy but also radiotherapy (thought nuclear DNA represents the most important target for IR), it would be interesting to add a paragraph or a short chapter on the effect of IR on these organelles and consequently redox signalling.
The review also mentions methods usable to explore the reviewed topic. If this information is available, it could be nice to overview also some cancer types / permanent cancer cell cultures that suffer from increased/decreased ROS levels and how this is related to mitochondria morphology/functioning and apoptosis activation.
I have checked "Accept after minor revision" to provide the authors with time to consider my notes. However, the above points should be considered as recommendations and the manuscript may be, in principle, published in the present form.
Author Response
What I miss i a schematic diagram showing mutual (and other important interactions) between proteins discussed in the text and involved in mitochondria "management", redox homeostasis and apoptosis. It would dramatically improve readability of the text.
A new schematic depicting the molecular players responsible for apoptosis sensitization via modulating mitochondrial fission has been presented as Figure 3 (line 488).
If the manuscript becomes too long, the chapter about the apoptosis (that mostly summarizes the "textbook" knowledge) may be reduced.
The section on apoptosis has been shortened (lines 202–215 deleted from the old version).
As mitochondria are not only targeted by chemotherapy but also radiotherapy (thought nuclear DNA represents the most important target for IR), it would be interesting to add a paragraph or a short chapter on the effect of IR on these organelles and consequently redox signalling.
We have included a short statement as an example of the effect of ionizing radiation has on mitochondrial dynamics and ROS production as part of a new section “2.2. The Crosstalk between Mitochondrial Shape Changes and Reactive Oxygen Species” (116).
The review also mentions methods usable to explore the reviewed topic. If this information is available, it could be nice to overview also some cancer types / permanent cancer cell cultures that suffer from increased/decreased ROS levels and how this is related to mitochondria morphology/functioning and apoptosis activation.
A short list of tumors displaying aberrant ROS production have been incorporated into the section 3.4 (451).